# Quorums enable optimal pooling of independent judgements in biological systems

James AR Marshall[1]*, Ralf HJM Kurvers[2], Jens Krause[3], Max Wolf[3]*

[1]Department of Computer Science, University of Sheffield, Sheffield, United Kingdom; [2]Centre for Adaptive Rationality, Max Planck Institute for Human Development, Berlin, Germany; [3]Department of Fish Behavior and Ecology, Leibniz-Institute of Freshwater Ecology and Inland Fisheries, Berlin, Germany

**Abstract** Collective decision-making is ubiquitous, and majority-voting and the Condorcet Jury Theorem pervade thinking about collective decision-making. Thus, it is typically assumed that majority-voting is the best possible decision mechanism, and that scenarios exist where individually-weak decision-makers should not pool information. Condorcet and its applications implicitly assume that only one kind of error can be made, yet signal detection theory shows two kinds of errors exist, 'false positives' and 'false negatives'. We apply signal detection theory to collective decision-making to show that majority voting is frequently sub-optimal, and can be optimally replaced by quorum decision-making. While quorums have been proposed to resolve within-group conflicts, or manage speed-accuracy trade-offs, our analysis applies to groups with aligned interests undertaking single-shot decisions. Our results help explain the ubiquity of quorum decision-making in nature, relate the use of sub- and super-majority quorums to decision ecology, and may inform the design of artificial decision-making systems.

**Editorial note:** This article has been through an editorial process in which the authors decide how to respond to the issues raised during peer review. The Reviewing Editor's assessment is that all the issues have been addressed (see decision letter).

DOI: https://doi.org/10.7554/eLife.40368.001

*For correspondence:
james.marshall@sheffield.ac.uk
(JARM);
m.wolf@igb-berlin.de (MW)

**Competing interests:** The authors declare that no competing interests exist.

## Introduction

Effective decision-making is essential in all areas of human society and, more generally, for all organisms. A fundamental question in this context is when a group of decision-makers is superior to individual decision-makers and vice-versa (*Galton, 1907*; *Surowiecki, 2005*; *Bahrami et al., 2010*; *Lorenz et al., 2011*; *Koriat, 2012*; *Kurvers et al., 2016*). Both in human and animal collective decision-making, the Condorcet Jury Theorem is one of the key principles guiding our thinking about this question (*List, 2004*; *Hastie and Kameda, 2005*; *King and Cowlishaw, 2007*; *Sumpter et al., 2008*; *Austen-Smith and Feddersen, 2009*; *Conradt and List, 2009*; *Kao and Couzin, 2014a*; *Marshall et al., 2017*). In a nutshell, for pairwise decision problems (e.g. disease-, lie- or predator detection), Condorcet's Jury Theorem states that a group of decision-makers employing the majority rule is superior to individual decision-makers in contexts where individuals are relatively accurate (i.e. accuracy >50%); conversely, individual decision-makers are superior to majority voting groups in contexts where individuals are relatively inaccurate (i.e. accuracy <50%). Consequently, across diverse fields ranging from organismal behaviour and human psychology to political sciences, two heuristics are commonly employed (*List, 2004*; *Hastie and Kameda, 2005*; *King and Cowlishaw, 2007*; *Sumpter et al., 2008*; *Conradt and List, 2009*; *Kao and Couzin, 2014a*): (i) groups of decision-makers outperform individuals only in contexts where individuals are relatively accurate and (ii)

the majority rule is a powerful mechanism to reap the benefits of collective decision-making. We here show that both statements are not true, and in doing so explain the ubiquity of quorum decision rules in the collective behaviour of humans and other social organisms (*Seeley and Visscher, 2004*; *Sumpter and Pratt, 2009*; *Ward et al., 2012*; *Pratt et al., 2002*; *Ross-Gillespie and Kümmerli, 2014*; *Walker et al., 2017*; *Bousquet et al., 2011*).

Over the past few decades, substantial research effort has focussed on two key explicit assumptions underlying Condorcet's Jury Theorem, independence (i.e. judgments/votes by different members of the group are assumed to be independent from each other) and homogeneity (i.e. all decision-makers within a group are assumed to be identical, both in competence and in goals) (*Kao and Couzin, 2014a*; *Sumpter and Pratt, 2009*; *Boland, 1989*; *Ladha, 1992*; *Berg, 1993*; *Marshall et al., 2017*). We here focus on a third, implicit, assumption of Condorcet's Jury Theorem, namely that decision-makers make only one type of error. This assumption stands in contradiction to the well-known fact that, when confronted with a pairwise decision problem like the one studied in Condorcet's Jury Theorem, two different types of error are possible (i.e. false positive and false negative). Surprisingly, up to now, this basic and well-known feature of pairwise decision problems has not been fully taken into account when investigating Condorcet's Jury Theorem.

We start by providing a brief summary of the basic model considered in Condorcet's Jury Theorem. We then introduce an extended model that takes into account the fact that decision-makers can make two types of errors. Based on this extended model, we then show that Condorcet's Jury Theorem makes several important predictive errors, which apply to the majority of decision scenarios. Moreover, majority voting is frequently suboptimal, whereas quorum-based voting with an appropriate quorum is always optimal, in that it enables groups to simultaneously maximise true positive rate and minimise false positive rate (*Wolf et al., 2013*). While an analytical solution for the optimal quorum threshold has been derived before (*Ben-Yashar and Nitzan, 1997*), dependent on true and false positive rates and key ecological characteristics (i.e. classification error cost, prior probabilities), this analysis treated true and false positive rates as independent of these ecological characteristics, whereas in reality the former depend on the latter. In contrast, here we also make use of signal detection theory to optimise individual decision-makers, thereby delineating precisely where non-majority quorums should be used, depending on parameters of the decision ecology. Thus, the simple majority threshold is only a special case of the more general quorum decision mechanism, in which optimal sub-majority or super-majority quorums are the rule rather than the exception.

## Methods

### Condorcet's Jury Theorem: the basic model

Condorcet's Jury Theorem considers a binary (pairwise) choice situation, in which a decision-maker can choose between two actions, labelled +1 and −1. Each decision-maker is characterised by a single parameter $a$, corresponding to the probability of making a correct decision, or decision accuracy. Importantly, the decision accuracy $a$ of each decision-maker is assumed to be conditionally-independent of the realised decisions of all other decision-makers.

Condorcet's Jury Theorem now considers a group of identical decision-makers of size $N$ that performs a majority vote. A simple combinatorial argument shows that if the accuracy of decision-makers is above 50% (i.e. $a > 0.5$), then the probability of making a correct choice increases with increasing group size and asymptotically approaches 1 (*Boland, 1989*). Conversely, if the accuracy of individual decision-makers is below 50% (i.e. $a < 0.5$), then the probability of making a correct choice decreases with increasing group size and asymptotically approaches 0 (*King and Cowlishaw, 2007*). This is because, as group size increases, the probability of the more probable decision (+ or −) also being the majority decision rapidly increases towards one.

These results have led to three key interpretations: first, pooling independent judgements is beneficial (i.e. improves decision accuracy) whenever individuals are relatively good decision-makers ($a > 0.5$) (*List, 2004*; *King and Cowlishaw, 2007*; *Novaes Tump et al., 2018*). Second, pooling judgements is detrimental (i.e. decreases decision accuracy) whenever individuals are poor decision-makers ($a < 0.5$) (*List, 2004*; *King and Cowlishaw, 2007*; *Novaes Tump et al., 2018*). Third, the

majority rule is the appropriate mechanism to reap the benefits of collective decision-making (*List, 2004*; *Hastie and Kameda, 2005*).

In the following, we show that each of these interpretations is incorrect. In particular, we show that (i) in cases where decision-makers are good ($a > 0.5$) majority decisions may decrease decision accuracy, (ii) in cases where decision-makers are poor ($a < 0.5$) majority decisions may increase decision accuracy, and (iii) pooling independent decisions is always beneficial as long as an appropriate quorum-based decision rule is used. Taken together, these results show that, for a large proportion of decision scenarios, the simple majority decision rule performs poorly and gives incorrect predictions about group decision accuracy.

## Condorcet's Jury Theorem: an extended model

Condorcet's Jury Theorem implicitly assumes that decision-makers can make only one error, that is, the probability of making an incorrect decision is $1 - a$. This is in contrast to the well-known fact that – in pairwise decision problems – decision-makers can make two types of errors, false positives and false negatives (*Green and Swets, 1966*; *Swets, 1988*). For example, an animal under predation risk may run away in the absence of a predator (false positive) or it may not run away in the presence of a predator (false negative) (*Trimmer et al., 2008*). Similarly, a doctor screening for a disease may diagnose in the absence of a disease (false positive) or not diagnose in the presence of a disease (false negative) (*Wolf et al., 2013*). As we discuss below, the implicit assumption of Condorcet's Jury Theorem is equivalent to assuming that decision-makers have an identical probability of committing the two errors – this is an important assumption that does not reflect the vast majority of real world decisions.

We here consider an extension of the above-described model, which takes into account the fact that decision-makers can make two types of errors. Again, decision-makers can choose between two actions, +1 and –1. Unlike in the basic model above, however, and consistent with standard decision theory for pairwise decisions, we now assume that the world can be in two states, state + and state –, corresponding to, for example, the presence and absence of a predator or the presence and the absence of a disease; state + holds with probability $p$ and state – thus holds with probability $1 - p$. Action +1 is the better choice in state + (i.e. it achieves a higher payoff), while action –1 is the better choice in state –; for example, running away is better than staying in the presence of a predator, while staying is better than running away in the absence of a predator. Consequently, and in contrast to the basic Condorcet model above, each decision-maker is now characterised by two parameters $a_+$ and $a_-$, corresponding to the probabilities of making correct decisions in state + and state –, respectively; this inevitably implies that individuals make two types of errors. As in the simple model above, where decision-makers have equal accuracies $a$ and are independent, for any individual in a group of $N$ individuals the probability of making a correct decision based on the true state of the world is conditionally-independent of the probabilities of other group members making the correct decision, given that same state of the world.

To relate our analysis to predictions made by applying Condorcet, we must define expected individual accuracy $a$, as used in Condorcet, in terms of our state probability $p$ and state-wise accuracy parameters $a_+$ and $a_-$. The expected individual accuracy is thus

$$a = p a_+ + (1-p) a_-. \tag{1}$$

From *Equation 1* we can see that Condorcet implicitly assumes that accuracies in the two states of the world are equal since then $p$ disappears from the equation; as shown in Supplementary Information, this occurs when both states of the world are equally likely, and the costs and benefits of classifications in the two states of the world are equal, although asymmetric decision problems can also result in equal accuracies (as can be confirmed with reference to Supplementary Information for *Figure 1*). Later, we explain how signal detection theory determines optimal values of $a_+$ and $a_-$ for a given decision scenario where these assumptions are violated.

## Results

We are now ready to formalise when Condorcet gives incorrect predictions, and when simple majority voting is suboptimal. We begin by determining optimal values of $a_+$ and $a_-$. Given the prior

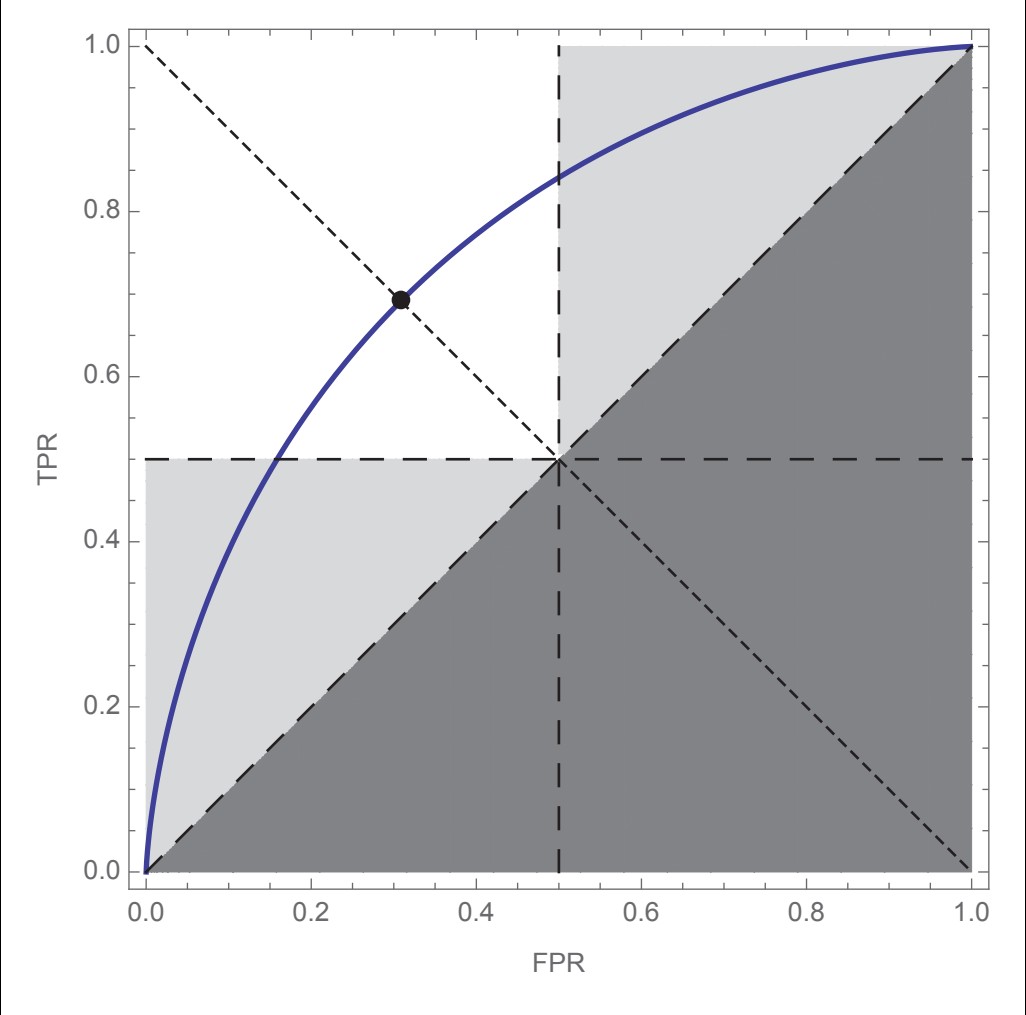

**Figure 1.** Receiver Operating Characteristic (ROC) curve plot showing the optimal compromise between true positive rate (TPR = $a_+$) and false positive rate (FPR = $1 - a_-$). The shape of the ROC curve is determined by the difficulty of the decision problem (e.g. *Shettleworth, 2010*), with hard decision curves lying close to the dashed diagonal, and easy decision curves approaching the left and top edge of the plot. According to prior probability of states, and relative cost of errors, an optimal decision-maker then selects a point on the ROC curve that gives the best possible expected decision performance; points that do not prioritise accuracy in either state of the world ($a_+ = a_- \Leftrightarrow$ TPR = 1 − FPR; dotted diagonal, and filled circle) are implicitly assumed by the Condorcet Jury Theorem, as discussed in the main text and *Figure 2*. The white square represents the region of ROC space in which simple majority decisions are best, and Condorcet predictions are fulfilled (note the equal accuracy dotted line described above always occupies this region). Light grey triangles represent regions of ROC-space which, if selected by optimal individual decision-makers, lead to sub-optimal collective decisions when combined by simple majority decision rules; in this region Condorcet predictions are systematically incorrect. The dark grey lower-right triangle represents combinations of TPR and FPR that should not be observed, since in these decision-makers are systematically wrong and could simultaneously improve both their TPR and FPR by simply inverting their predictions to move above the dashed diagonal. ROC parameters: $\mu_- = 0$, $\mu_+ = 1$, $\sigma = 1$, prior = ½, ratio = 1.
DOI: https://doi.org/10.7554/eLife.40368.002

probability $p$, the cost and benefits associated with the two states of the world, and the state-dependent characteristics of the cue(s) decision-makers base their decisions on, the optimal realised individual accuracies $a_+$ and $a_-$ are derived by solving a signal detection problem (*Figure 1*). Signal detection theory has been applied repeatedly to hypothetical and real world decision problems, for example predator detection by foraging animals (*Trimmer et al., 2008*; *Trimmer et al., 2017*); diagnostic decision-making (*Kurvers et al., 2016*; *Wolf et al., 2015*); and lie detection (*Klein and Epley,*

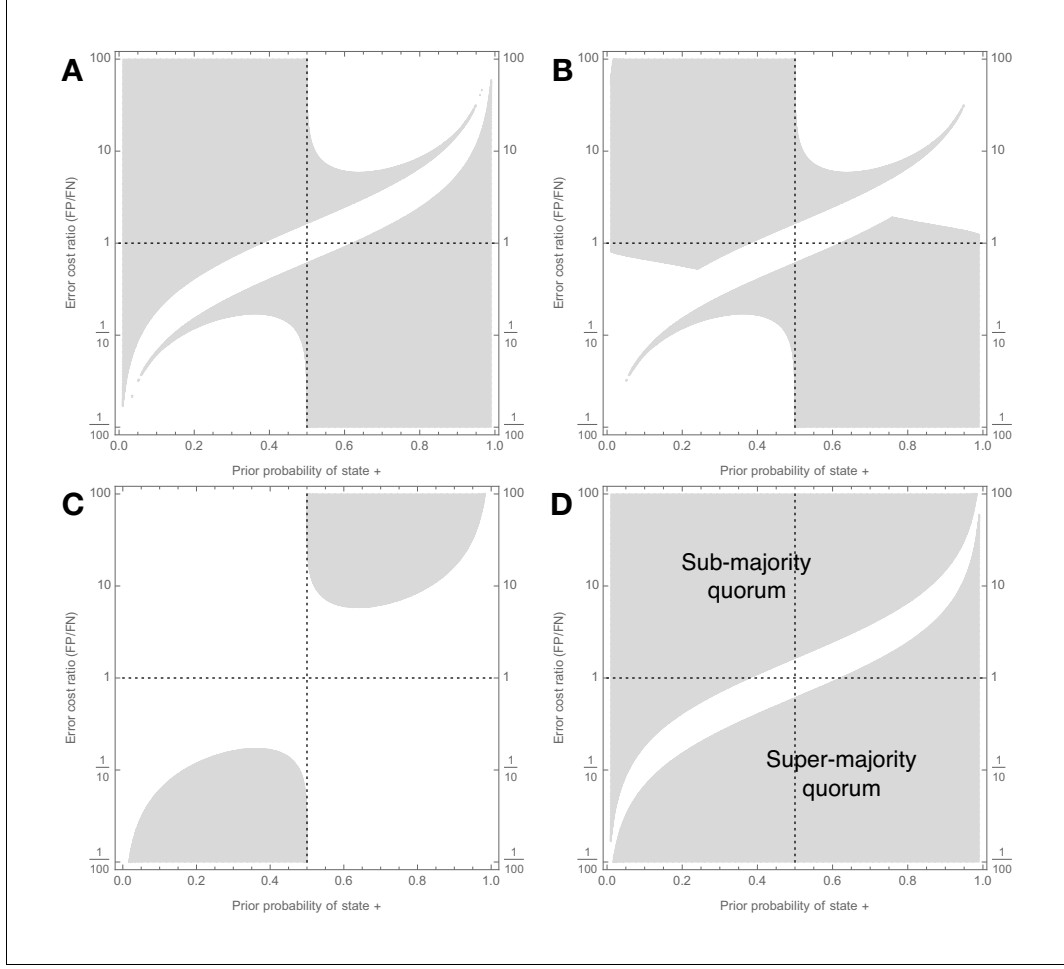

**Figure 2.** Signal detection theory shows when majority-based reasoning is incorrect. Decision scenarios in which Condorcet will make different kinds of inaccurate predictions about groups of individually-optimal (see *Figure 1*) decision-makers (a–c), and in which majority voting is sub-optimal (d); prior probability of state + is on x-axis, ratio of cost of false positive to cost of false negative is on y-axis, so values larger than one indicate classifying – as + is relatively worse than classifying + as –, and vice versa. Grey regions show decision scenarios in which Condorcet leads to at least one predictive error: (a) As group size increases Condorcet incorrectly predicts that majority group accuracy will asymptotically increase towards 1, whereas it actually does not (Error Ia, main text); (b) As group size increases Condorcet incorrectly predicts that majority group accuracy will asymptotically increase towards 1, whereas groups actually make worse decisions than individuals (Error Ib, main text); (c) As group size increases Condorcet incorrectly predicts that majority group accuracy will decrease towards zero, whereas it actually converges to a positive level (Error II, main text); (d) Condorcet makes at least one of the errors just described; this plot also corresponds to decision scenarios in which majority decision-making is suboptimal, and should be replaced by a sub- (upper-left area) or super- (lower-right area) majority quorum rule as described in Results. Signal detection analysis for individual decision-makers is described in the Appendix; parameters for the analysis are $\mu_- = 0, \mu_+ = \sigma = 1,\ C_{TP} = C_{TN} = 0,\ C_{FN} = 1,\ C_{FP} = \text{ratio}$.

DOI: https://doi.org/10.7554/eLife.40368.003

2015). As described in the appendix, for simplicity and tractability our analysis is conducted for the simplest signal detection problem, determining which of two normal distributions a single scalar random variable is drawn from; this could, in the example of a predator detection problem, be the instantaneous volume of a sound heard by a forager. In this case signal detection theory enables us to find a Receiver Operating Characteristic (ROC) curve of optimal $a_+, a_-$ value pairs, dependent on decision difficulty (*Figure 1*; *Green and Swets, 1966*; *Shettleworth, 2010*); the optimal point of

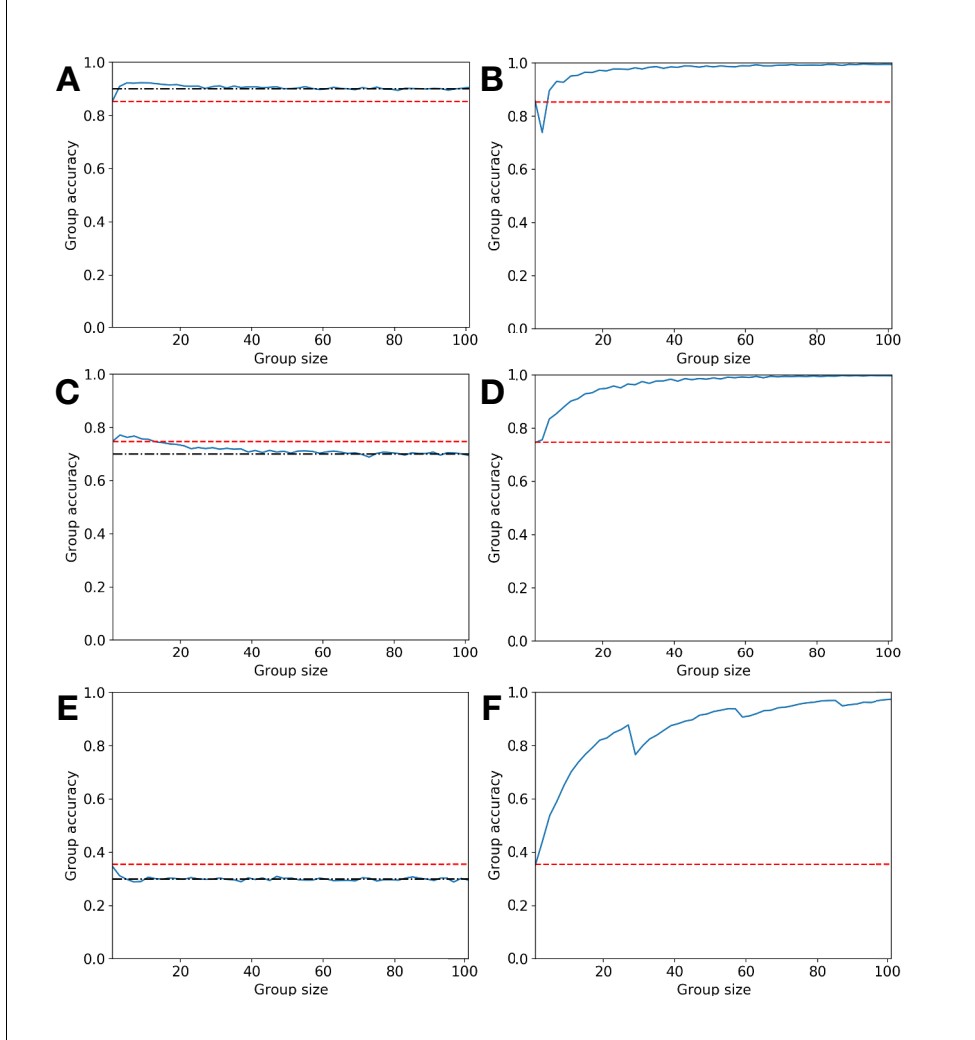

**Figure 3.** How majority voting can be optimally replaced by quorum decisions. Illustrative failures of simple majority voting and Condorcet predictions (left column) and their remediation through appropriate use of sub- or super-majority decision quorums (right column); group size increases on x-axis, while group accuracy increases on y-axis; red-dashed line indicates individual decision accuracy, black dot-dash line represents prior probability of state + (panels a and c) or of state − (panel e), solid blue line represents group accuracy. (a) Error Ia: Condorcet predicts that increasing group size will result in group accuracy converging to 1, but it converges to $p$ which is above the individual accuracy level. (b) Choosing a super-majority quorum leads group accuracy to converge to 1 with increasing group size. (c) Error Ib: Condorcet predicts that increasing group size will result in group accuracy converging to 1, but it converges to $p$ which is below the individual accuracy level, hence groups perform worse than individuals. (d) Choosing a super-majority quorum leads group accuracy to converge to 1 with increasing group size. (e) Error II: Condorcet predicts that increasing group size will result in group accuracy converging to 0, but it converges to $1 - p$ which is below the individual accuracy level. (f) Choosing a sub-majority quorum leads group accuracy to converge to 1 with increasing group size. Simulations comprise 10,000 replicates with individual accuracies derived from signal detection analysis with parameters $\mu_- = 0, \mu_+ = \sigma = 1, \ C_{TP} = C_{TN} = 0, \ C_{FN} = 1$ and: (a) $C_{FP} = 4, p = 0.9, q = 0.5$, (b) $C_{FP} = 4, p = 0.9, q = 0.7$, (c) $C_{FP} = 1, p = 0.7, q = 0.5$, (d) $C_{FP} = 1, p = 0.7, q = 0.75$, (e) $C_{FP} = 10, p = 0.7, q = 0.5$, (f) $C_{FP} = 10, p = 0.7, q = 0.035$.

DOI: https://doi.org/10.7554/eLife.40368.004

this curve gives us a unique pair of $a_+$ and $a_-$ values dependent on costs and benefits of different decision outcomes, and the prior probability of the + state, $p$.

We next make the observation that, to simultaneously improve both true positive and false positive rates, a group must choose a quorum $q$ that lies between these two values (**Wolf et al., 2013**); that is, an optimal group must choose $q$ such that

$$1 - a_- < q < a_+ \qquad (2)$$

The intuition behind this result is that group accuracy will converge on the appropriate accuracy for the true state of the world, as group size increases; thus, by setting a quorum between these two accuracies the true state of the world can be determined with high probability for sufficiently large groups. For further details see (**Wolf et al., 2013**). Since the simple majority quorum $q = 1/2$, assumed by Condorcet, only satisfies this inequality when both $a_+ > 1/2$ and $a_- > 1/2$, when either of these conditions are violated then both simple majority decisions and Condorcet-based reasoning will be deficient. Furthermore, we should never see true positive rate ($a_+$) less than false positive rate ($1 - a_-$) (dark grey region of ROC space in **Figure 1**), since such a decision-maker could simultaneously improve both their true and false positive rates simply by inverting their decisions and moving above the diagonal. Therefore, the ROC space is divided into two meaningful regions: in the first $a_+ > 1/2$ and $a_- < 1/2$ (white region in **Figure 1**), so simple majority voting is asymptotically-optimal as group size increases, and Condorcet-based predictions are correct. In the second, $a_+ < 1/2$ and $a_- > 1/2$, or $a_+ > 1/2$ and $a_- < 1/2$, while ensuring $a_+ > 1 - a_-$ (light grey regions in **Figure 1**); in these regions simple majority decisions will be sub-optimal, and Condorcet-based reasoning will be erroneous.

We now describe the systematic errors Condorcet leads to when faced with these decision scenarios. These errors can also be described in terms of false positive (frequently referred to as type I) and false negative (type II) errors, in predicting the performance of the Condorcet majority rule, hence we label the errors in such terms.

## Error Ia: Condorcet predicts group accuracy approaches 1, but majority groups do not

This error occurs when majority voting is sub-optimal ($a_+ < 1/2$ and $a_- > 1/2$, or $a_+ > 1/2$ and $a_- < 1/2$; light grey regions in **Figure 1**, as described above), and when expected individual accuracy $a > 1/2$, since under this last condition individuals on average make more correct decisions than incorrect decisions, and Condorcet thus predicts that group accuracy approaches one as group size increases (**King and Cowlishaw, 2007**; **Boland, 1989**). From **Equation 1** this requirement is thus that

$$p(a_+ - a_-) + a_- > 1/2 \qquad (3)$$

While Condorcet predicts group accuracy (denoted $\bar{a}$) approaches 1, that is

$$\lim_{n \to \infty} \bar{a} = 1 \qquad (4)$$

in fact the majority quorum $q = 1/2$ is either below $1 - a_-$, or above $a_+$, *contra* inequality 2; thus the group converges to making completely correct choices in one state of the world, and completely incorrect choices in the other state of the world (**Wolf et al., 2013**). Hence, group accuracy converges to

$$\lim_{n \to \infty} \bar{a} = \begin{cases} p & \text{if } a_+ > \frac{1}{2} \text{ and } a_- < \frac{1}{2} \\ 1 - p & \text{if } a_+ < \frac{1}{2} \text{ and } a_- > \frac{1}{2} \end{cases} \qquad (5)$$

The wide range of decision scenarios in which Condorcet makes this predictive error are illustrated in **Figure 2a**. An example of this error is presented in **Figure 3a**; **Figure 3b** illustrates how the error can be avoided by choosing an appropriate quorum, in this case a super-majority quorum.

## Error Ib: Condorcet predicts group accuracy approaches 1, but majority groups are worse than individuals

In error Ia, while group accuracy does not converge to one with increasing group size, nothing is said about whether or not groups are better than individuals. Error Ib refines error Ia, by showing

that there are cases where Condorcet predicts group accuracy approaching 1, but groups actually have lower accuracy than individuals. These cases are found by refining the conditions given in error Ia to include the additional condition that the group accuracy converged to (*Equation 5* above) is less than individual expected accuracy (*Equation 1*). This gives the conditions

$$p < \frac{a_-}{1 - a_+ + a_-} \text{ if } a_+ > \frac{1}{2} \text{ and } a_- < \frac{1}{2}, \text{ and} \tag{6}$$

$$p > \frac{1 - a_-}{1 + a_+ - a_-} \text{ if } a_+ < \frac{1}{2} \text{ and } a_- > \frac{1}{2} \tag{7}$$

The wide range of decision scenarios in which Condorcet makes this predictive error are illustrated in *Figure 2b*. An example of this error is presented in *Figure 3c*; *Figure 3d* illustrates how the error can be avoided by choosing an appropriate quorum, in this case a super-majority quorum.

## Error II: Condorcet predicts group accuracy approaches 0, but majority groups do not

In error II individual expected accuracy is below 1/2, thus Condorcet predicts that group accuracy should converge to 0; while groups using the majority decision rule do decrease in accuracy, they converge to a non-zero group accuracy given by *Equation 5* above. To find cases where this occurs we simply solve for when individual expected accuracy $a < 1/2$. From *Equation 1* this gives us the conditions

$$p < \frac{1/2 - a_-}{a_+ - a_-} \text{ if } a_+ > \frac{1}{2} \text{ and } a_- < \frac{1}{2}, \text{ and} \tag{8}$$

$$p > \frac{1/2 - a_-}{a_+ - a_-} \text{ if } a_+ < \frac{1}{2} \text{ and } a_- > \frac{1}{2} \tag{9}$$

Decision scenarios in which Condorcet makes this predictive error are illustrated in *Figure 2c*. An example of this error is presented in *Figure 3e*; *Figure 3f* illustrates how the error can be avoided by choosing an appropriate quorum, in this case a sub-majority quorum.

Note that it is not possible to find an 'Error IIb' case that parallels Error Ib; that is if Condorcet predicts that group accuracy approaches 0, majority groups will always be worse than individuals and never better, even if their group accuracy remains positive. This is because it is not possible simultaneously to satisfy the conditions just given (inequalities 8 and 9), and the opposite of the conditions (inequalities 6 and 7) given in *Error Ib* (i.e. the conditions that group accuracy converged to is *greater* than individual expected accuracy), as can be confirmed by algebra.

## Majority voting is usually suboptimal

Combining the cases in which Condorcet makes one of the above described predictive errors, *Figure 2d* illustrates when Condorcet will make at least one error in predicting the performance of decision-making groups using majority decision-making. This set also corresponds to the set of decision scenarios in which majority decision-making is sub-optimal, and is outperformed by an appropriately set sub- or super-majority quorum. *Figure 2d* shows that Condorcet is optimal in far fewer decision scenarios than those in which it is outperformed by an appropriate quorum rule. Furthermore, our analysis also relates decision-ecology to the requirement for sub- or super-majority quorums. From inequality (2), sub-majority quorums are required whenever $a_+ < 1/2$, which corresponds to the upper-left area of *Figure 2d*; in contrast, super-majority quorums are required whenever $1 - a_+ > 1/2$, which corresponds with the lower-right area of *Figure 2d* (see Supplementary Information). Thus, whenever the positive state of the world + is rarer, and/or false positives are relatively expensive compared to false negatives, then a sub-majority quorum should typically be employed, while the converse holds for super-majority quorums.

## Discussion

We have shown that simple majority-based collective decisions are often suboptimal, and that consequently sub- or super-majority quorums should frequently be employed by groups of 'like-minded' individuals combining independent decisions. Our results are important for two reasons. First, the majority-based decision rule, and Condorcet-based reasoning, is widespread in several major branches of collective decision theory. In the animal behaviour community the Condorcet prescription on individual accuracy exceeding ½ has been used to recommend when opinions should be pooled (*King and Cowlishaw, 2007*), and when experts should be favoured over group opinions (*Katsikopoulos and King, 2010*). Other authors have also invoked Condorcet and majority voting as the gold-standard for collective decision-making (*Hastie and Kameda, 2005*; *Kao and Couzin, 2014a*; *Kao et al., 2014b*; *Miller et al., 2013*). These results have been useful in highlighting the benefits of information pooling in collective decisions, but such studies implicitly neglect the reality that most decisions have two types of errors. Second, while quorums have been widely studied in collective decision theory, we here present a comprehensive theory that may help explain their prevalence. Sub- and super-majority quorums have been considered theoretically (*Sumpter and Pratt, 2009*); Sumpter and Pratt implicitly assume only one type of error need be considered, and proceed from that point with their analysis, referring to quorum functions such as managing speed-accuracy trade-offs (*Marshall et al., 2009*). *Ward et al. (2008)* consider quorum use for facilitating information transfer in shoaling fish, yet ignore the possibility of different error types despite their great asymmetry in the scenario studied, predator detection. *Conradt and Roper (2005)* in their review refer to true and false positives, but in explaining quorum usage refer back to earlier analysis as a mechanism to avoid extreme group decisions where individual fitness interests do not completely align (*Conradt and Roper, 2003*). However *List (2004)* as well as *Conradt and List (2009)* noted the effect of cost and prior asymmetry on quorum usage, referring back to earlier political science results (*Ben-Yashar and Nitzan, 1997*) discussed below. We also note that convergence on an intermediate group accuracy between 0 and 1 has previously been observed, without consideration of signal detection theory; the result presented in Figure 1b of *Kao and Couzin (2014a)* occurs for a similar reason to our error Ia, in that inappropriate use of a majority decision rule leads increasing group size to result in group accuracy converging on a parameter of the decision problem, in their case the reliability of a low correlation cue.

In contrast to these previous analyses here we have shown the fundamental importance of quorums in one of the simplest possible collective behaviour scenarios, single-shot collective decisions in homogenous groups, where individuals' interests are aligned, and decision-making abilities do not differ. Thus, we might expect the use of quorums to be widespread in the natural world, even in the simplest of decisions. The widespread use of quorum sensing in bacteria provides evidence of this (*Gross, 2017*), and may prove a particularly good testbed for our theory given its binary nature and asymmetric state priors and error costs, although evolutionary conflicts of interests within bacterial communities may result in confounds. Moreover, as humans have been shown to employ quorum rules and adaptively adjust the associated quorum thresholds, human decision-making experiments may also provide a powerful approach to test our predictions (*Kurvers et al., 2014*; *Clément et al., 2015*).

It is surprising that signal detection theory has seen relatively scant application to collective decision-making. *Wolf et al. (2013)* noted the potential for different error types in identifying how quorums can improve collective decision accuracy; while motivated by signal detection theory they did not directly apply it to optimise the individual decision-makers and relate this back to group behaviour. *Kirstein and von Wangenheim, 2010* also noted the possibility for independent error types, again with reference to signal detection theory; they noted the potential for Condorcet to make the same qualitatively incorrect predictions that we note here, however they did not apply the relevant theory to delineate the situations under which Condorcet reasoning is incorrect, nor did they consider the possibility for quorum-based decision rules to rescue these situations. *Sorkin et al. (2001)* applied signal detection theory to Condorcet-like models with varying super-majority quorums, but did not find the mechanism by which group decisions can be optimised (*Wolf et al., 2013*). *Laan et al. (2017)* noted that Condorcet and majority voting can be suboptimal, and considered ways in which voting can be improved; while they discussed signal detection theory and voting mechanisms they neither explicitly considered error types, nor quorum thresholds, focussing mainly

on correlations between decision-makers and the impact of the cost function used, as well as suggesting a data-driven machine learning approach to improving collective decision-making rules. The results that most closely anticipate ours are those of *Ben-Yashar and Nitzan (1997)*, who analytically solved for the general optimal decision rule by recognising both error cost and prior asymmetry, as well as simultaneously considering the case of variable individual decision-ability (e.g. *Marshall et al., 2017*). These results describe the relationship between prior asymmetry and optimal quorum threshold, and error cost and optimal quorum threshold. However, because they did not apply signal detection theory to optimise individual agents' decisions (i.e. they treated true and false positive rates of individuals as independent from the ecological parameters error cost and prior asymmetry), they were unable quantitatively to uncover the complex nonlinear relationship between these three quantities (*Figure 2*).

In contrast to earlier work, by applying signal detection theory we have simultaneously shown here both how fragile Condorcet and majority-voting are, and how the use of sub- or super-majority quorums should relate to decision ecology. Although simple collective behaviour models have been well studied and highly influential, our results, and others relaxing other assumptions of such models (*Marshall et al., 2017*), indicate the subtlety that may be revealed in collective decision-making by a richer consideration of individual decision theory. Other approaches to such problems should be pursued in the future. For example, it is possible to optimise individual quorums (*Ben-Yashar and Nitzan, 1997*) rather than simply set them within a suitable interval as we do here, thereby giving greater benefits to smaller groups. Similarly, rather than apply signal detection theory one could apply Bayesian decision-theory (e.g. *Pérez-Escudero and de Polavieja, 2011*; *Arganda et al., 2012*; *Pérez-Escudero and de Polavieja, 2017*), thereby attempting to deal with further complexities such as non-independence of individual decisions. In many scenarios errors and correct decisions may be correlated although even when multiple individuals observe the same stimulus their information can be considered independent due to sensory noise (*Marshall et al., 2017*). We believe that our simple approach has, however, the benefit of tractability while still revealing the complexity of collective decision-making even in the simplified scenario considered.

Our work takes inspiration from political science and decision theory to address questions in behavioural ecology, but may additionally have the potential to inform work in the design of artificial decision-making systems, machine learning and robotics. For example, in the field of ensemble learning, in which predictions from multiple weak classifiers such as neural networks are combined to improve decision accuracy, variable quorums, referred to as 'threshold shift', are used (e.g. *Dmochowski et al., 2010*). However majority voting is still among the simplest and most ubiquitous vote fusion rules discussed (*Sagi and Rokach, 2018*; *Krawczyk et al., 2017*). Hence, we suggest that the simple perspective on how to combine votes presented here may also yield technological insight.

## Acknowledgements

We thank Gavin Brown and Nikolaos Nikolaou for helpful discussions on ensemble learning theory, and Andrew King, Gonzalo de Polavieja and an anonymous reviewer for helpful comments during the review process. JARM was funded by the European Research Council (ERC) under the European Union's Horizon 2020 research and innovation programme (grant agreement number 647704).

## Additional information

### Funding

| Funder | Grant reference number | Author |
|---|---|---|
| H2020 European Research Council | 647704 | James AR Marshall |

The funders had no role in study design, data collection and interpretation, or the decision to submit the work for publication.

#### Author contributions
James AR Marshall, Max Wolf, Conceptualization, Formal analysis, Writing—original draft, Writing—review and editing; Ralf HJM Kurvers, Jens Krause, Conceptualization, Writing—review and editing

#### Author ORCIDs
James AR Marshall (iD) http://orcid.org/0000-0002-1506-167X
Ralf HJM Kurvers (iD) https://orcid.org/0000-0002-3460-0392

#### Decision letter and Author response
Decision letter https://doi.org/10.7554/eLife.40368.011
Author response https://doi.org/10.7554/eLife.40368.012

## Additional files
#### Supplementary files
• Transparent reporting form
DOI: https://doi.org/10.7554/eLife.40368.005

• Source code 1. Source code for results presented in *Figure 1*.
DOI: https://doi.org/10.7554/eLife.40368.007

• Source code 2. Source code for results presented in *Figure 2*.
DOI: https://doi.org/10.7554/eLife.40368.008

• Source code 3. Source code for results presented in *Figure 3*.
DOI: https://doi.org/10.7554/eLife.40368.009

#### Data availability
All data generated or analysed during this study are included in the manuscript and supporting files. Source data files have been provided for Figures 1, 2 and 3.

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

## Appendix 1

DOI: https://doi.org/10.7554/eLife.40368.006

# Signal detection theory reveals cases where Condorcet Predictions are incorrect

To determine if a group of optimal decision-makers could exist such that simple application of Condorcet would lead to erroneous predictions, but a quorum rule would allow optimal opinion pooling, we consider groups of identically-capable individual decision-makers (as assumed in Condorcet), modelled as making optimal decisions using signal detection theory in order to classify continuous signals arising from one of two possible normal signal distributions of equal variance (**Green and Swets, 1966**). That is, decision-makers are faced with a signal

$$s \sim \begin{cases} N(\mu_+, \sigma) \text{ with probability } p \\ N(\mu_-, \sigma) \text{ with probability } 1-p \end{cases} \tag{A.1}$$

and must choose an optimal signal threshold, $x$, in order to classify signals as being drawn from either of the two possible normal distributions. Each distribution has a different mean ($\mu_+$ versus $\mu_-$) but the same standard deviation ($\sigma$). In a natural setting the signal could represent information as to whether a predator is present or not, for example, with the two states of the world, predator present versus predator absent, having different distributions for this signal.

The optimal decision threshold $x$ is chosen to minimise the expected loss for the decision-maker (or maximise the expected gain). The expected loss from a decision is

$$E(\text{Loss}) = p(a_+ C_{TP} + (1-a_+)C_{FN}) + (1-p)(a_- C_{TN} + (1-a_-)C_{FP}), \tag{A.2}$$

where $C_{TP}$, $C_{FN}$, $C_{TN}$ and $C_{FP}$ are respectively the costs of true positives (correctly classifying state +), false negatives (incorrectly classifying state –), true negatives (correctly classifying state –) and false positives (incorrectly classifying state +). Thus, an optimal decision-maker should minimise (**equation A.2**) by appropriately choosing the decision threshold $x$. Since, given $x$,

$$a_+ = 1 - \Phi(\mu_+, \sigma, x), \text{ and} \tag{A.3}$$

$$a_- = \Phi(\mu_-, \sigma, x), \tag{A.4}$$

where $\Phi$ is the cumulative distribution function for the normal distribution, the optimal $x$ can be found by substituting (**equation A.3**) and (**equation A.4**) into (**equation A.2**), differentiating the resulting equation and solving for zero (**Green and Swets, 1966**). Note that the optimal threshold $x$, and thus the optimal accuracies under state + and state – of the world, $a_+$ and $a_-$ respectively, are affected both by class imbalance ($p \neq 1/2$), and by asymmetric error costs ($C_{TP} - C_{FN} \neq C_{TN} - C_{FP}$).

