## [Decision Letter]

[**Editorial note:** This article has been through an editorial process in which the authors decide how to respond to the issues raised during peer review. The Reviewing Editor's assessment is that all the issues have been addressed.]

Thank you for submitting your article "Quorums enable optimal pooling of independent judgements" for consideration by *eLife*. Your article has been reviewed by two peer reviewers, and the evaluation has been overseen by a guest Reviewing Editor and Joshua Gold as the Senior Editor. One of the two reviewers has agreed to reveal his identity: Gonzalo G de Polavieja (Reviewer #2).

The Reviewing Editor has highlighted the concerns that require revision and/or responses, and we have included the separate reviews below for your consideration. If you have any questions, please do not hesitate to contact us.

Summary:

This is a well written and interesting contribution to our understanding of the mechanisms underlying collective decisions. The manuscript is set out to explain the error of our ways when using Condorcet's theory to inform our understanding, and present refinements of the model to broaden its applicability and use in biological and human sciences. I (and the two reviewers) support the publication of this work, but would all like to see the authors consider a number of points during the revision process. Below, I outline these main issues that have arisen both as a result of the initial reviews and our interactive discussion.

Concerns:

1) To "explain the error of our ways" is fine, expect some (if not most) of the theory and empirical work cited acknowledge that the Condorcet's theory represents a very simplified view of the world, and more complex models/mechanisms can be employed. We would like to see the authors 'soften' the critiques of earlier work so to better represent the advance being made here (e.g. paragraph 1 and 2, Discussion section) and ensure researchers in the field take forward and use the results presented.

2) Regarding earlier work, reviewer #1 points to an earlier model that presents a similar model and conclusion to that presented here (Ben-Yashar and Nitzan, 1997). As a behavioural ecologist who has interest in these models, I was unaware of it, and therefore many in the field may be in the same position. As the work is presented in a narrative form,… "first we did this, next we tested, etc." it would be useful to weave into the narrative how this work is similar or different to what is being presented, as explained nicely by reviewer #1. Moreover, the 1970s through 1990s saw a lot of theoretical extensions to Condorcet, and today, there's a lot of work studying the effect of social influence, correlations, heterogeneity, emergent sensing, etc. on collective wisdom, which are all deviations from Condorcet. This should be more explicitly acknowledged early in the manuscript.

3) Related to points 1 and 2, the paragraph setting out the three key interpretations (Introduction section) requires citations to back up each of the statements. I can think of one of my own papers that would be appropriate to cite, but how many more? The authors are suggesting we have all been misled, so we need to have a good number of citations here to back up this claim.

4) The definition of 'optimal' and its relation to accuracy in decision-making (here, related to probability of correct decision) needs to be explained properly in the Introduction, or else a different more representative term used (accurate/accuracy?).

5) Non-independence of errors is probably the rule rather than the exception in nature, as well as in lab experiments with animals or humans. Both reviewers query if and how this will effect either the construction of "state-of-the-art" models like this one, or their interpretation and generality – if not explicitly incorporated. At very least, this should be discussed.

6) Reviewer #2 thinks a Bayesian analysis would be as or more insightful than the signal detection theory, then it is likely so will others. Therefore, I would encourage Authors to present an argument against this in the discussion (where the use of signal detection theory is highlighted: paragraph three, Discussion section), and/or consider their recommendation to work out a general solution/expression using Bayesian analysis.

7) The Abstract should begin with one or two sentences giving a broader background. At present, it begins with very specific background regarding the Condorcet Jury Theorem. Also, the biological system should be indicated in the title and/or Abstract: please revise the title and/or Abstract with this advice in mind. A simple solution would be to could add "in biological systems" (or similar) to the title and then explain in the Abstract that the theory/models apply to all sorts of 'social' organisms.

Separate reviews (please respond to each point):

*Reviewer #1:*

In this manuscript, the authors draw attention to two ideas that are under-appreciated in the collective decision-making making literature: 1) that there are two errors that can be made in a binary decision (false positives and false negatives), rather than the single error (correct/incorrect), that is the focus of many previous studies, and 2) that there exists a family of collective decision rules, within which simple majority rule is a special case. The authors generalize Condorcet's jury theorem to allow for different false positive and false negative rates and show that for many scenarios, simple majority rule is not the optimal method to combine individual opinions. They identify three types of mostly different errors that can arise when naively applying the predictions of Condorcet's jury theorem, and show that an error will arise in the majority of parameter space. They demonstrate that a 'quorum' decision rule, defined as a threshold for making a decision is a fraction different from 50%, is optimal for these scenarios.

I agree with the authors that these two ideas need to be of greater theoretical and empirical research, and this manuscript does a good job of highlighting the importance of these ideas. The authors rightly show that in nature, the prior probabilities, and the costs/benefits, of different possible outcomes are generally not equal (there may rarely be a predator, but when there is one, the cost of being eaten is very high). They link the large literature on quorum decision-making to optimal decisions, demonstrating how quorum decisions may be optimal within the ecological scenarios in which species evolve, which is important.

My main concern with this manuscript is its novelty. In particular, there are two previously published papers (one which the authors cited and one which the authors did not), which together contain all of the main ideas and much of the results of the present manuscript. I describe these papers below, and their intersection with this manuscript, and then describe what I think are the novel sections remaining in this manuscript.

The first paper, which the authors did not cite, is: Ben-Yashar RC and Nitzan SI (1997) The optimal decision rule for fixed-size committees in dichotomous choice situations: the general result. International Economic Review, 38(1):175-186. The model in this paper incorporates the main ideas of the present manuscript, namely that the costs/benefits ('payoffs') may be different for different outcomes; the prior probability of different outcomes may be different; and the probabilities of false positives and false negatives may be different. The model described in Ben-Yashar and Nitzan, 1997, is essentially identical to the extended model described in the current manuscript. That paper also showed that 'quorum' rules are generally optimal, but rather than denote the region of parameter space in which this is the case, those authors went further and mathematically proved the exact value of the quorum threshold for each set of parameter values (they also further considered heterogeneous individuals, which was not considered in the current manuscript). In short, the optimal solution for the model appears to have been solved already.

The second paper, which the authors did cite, is: Kao and Couzin, 2014. This paper used a slightly different model but with some key similarities. In particular, the model described two cues in the environment, one which is independently perceived by each individual, and one which is globally perceived by all individuals. Although the authors did not describe the global cue as such, one interpretation of a global cue is as a prior probability of an outcome. By mapping the model in the current manuscript to the Kao and Couzin, 2014 model, one can find that the condition for "Error Ia" is the same as the boundary shown in Figure 1B of the Kao and Couzin paper. In the latter paper, the authors show that in this region of parameter space, the collective accuracy for very large groups approaches the accuracy of the global cue, which is the same result as Equation 5 and the left column of Figure 3 in the present manuscript.

To summarize, the model described in the present manuscript appears nearly identical to the one described in the Ben-Yashar and Nitzan paper, while the older paper gave a more general solution to than the present manuscript. Furthermore, one of the three errors presented in the present manuscript has been essentially described in the Kao and Couzin paper.

Despite all of this, I do think that there remain some novel results in the present manuscript. To my knowledge, Error Ib and Error II have not been previously described, although Error Ib is a subset of Error Ia. Furthermore, the way the authors parsed out the different errors and illustrated its prevalence in parameter space is useful. I also think that despite this model being solved already, the Ben-Yashar and Nitzan paper is not broadly known to the ecology/behavior community and drawing attention to it and its results is important. However, I think that the authors need to revise their manuscript in light of these two papers to highlight the aspects of their work that are truly novel.

Minor Comments:

I would suggest that the authors move most, or all, of the appendix to the main text, as it's not possible to understand the model or the results without reading this. For example, it's not clear why there is a trade-off between a+ and a- without knowing the details of the model (one may wonder why one can't simply set a+ and a- both to 1).

*Reviewer #2:*

The authors extend Condorcet analysis by taking into account false and negative errors instead of collapsing them into a single error. They work out the consequences of their analysis. I completely agree with the authors that the role of Condorcet theorem has been misinterpreted in the literature. It is a nice analytical model full of assumptions and we can learn a lot by realizing what is the role of these assumptions.

The author's key insight is that the world, for the agent, can be in different states, say + and -, with probabilities p and 1-p, and the error in each world state is different. In contrast, Condorcet's treatment is for a single world state.

There are two ideas I would consider:

1) I think a Bayesian analysis is actually as or more insightful than the signal detection theory. It naturally takes into account all types of errors and even correlations. You can also see in this treatment that you can do very well in supermajority or below majority cases.

2) In this respect, the analysis in the paper assumes errors to be independent. This is fine, Condorcet assumes independent voters. But this assumption does not hold in practice. So, again, I think the more general treatment is a Bayesian one including all types of errors and correlations (see Perez-Escudero and Polavieja, 2011 and more recently in Interface).

I think it would be nice to work it out from Bayes (maybe even more general using costs), obtain the general expression, and work out the signal-detection-theory case as a particular case.

As you are in this new experimental reviewing mode, you might try my suggestion (or not), and if it gets messy, leave it for another time (or not).

[Editors' note: further revisions were requested prior to acceptance, as described below.]

Thank you for resubmitting your work entitled "Quorums enable optimal pooling of independent judgements in biological systems" for further consideration at *eLife*. Your revised article has been favorably evaluated by Joshua Gold (Senior Editor), a guest Reviewing Editor, and one reviewer.

Thank you very much for the time taken to carefully consider the reviewers and my own comments and revise your manuscript. I sent your revised manuscript to reviewer 1 to consider the changes you have made. Reviewer 1 is still not convinced about the novelty of your manuscript, and has suggested further revisions in relation to the Ben-Yashar and Nitzan paper, which we have been discussing previously. In particular, they point out that an analytical solution for the optimal quorum threshold is provided in this study, and therefore, provide a quantitative rather than qualitative solution. I tend to agree, and think that it is important that the value of this work is not dismissed out-of-hand (especially since many will not take the time to go back and read it). Reviewer 1 suggests the Ben-Yashar and Nitzan paper deserves a more prominent role. You can and should follow this suggestion whilst emphasising your own advances (e.g., using signal detection theory).

Reviewer 1 also provides further details of a misunderstanding with respect to the Kao and Couzin model which you will need to address, since the description of this model provided by Kao and Couzin may well be closer to your own work than is immediately obvious from reading the original work (indeed, it was not clear to me). Please ensure that this aspect of the Kao and Couzin is cited and discussed properly.

Reviewer 1 comments:

1) Regarding the Ben-Yashar and Nitzan, 1997, paper, the authors write in their response letter: "…while they are able to make qualitative discussions of the relationships between asymmetries and the resulting changes in quorum threshold, they cannot quantify this, or reason about interactions," and in their revised manuscript, "These results describe a qualitative relationship between prior asymmetry and optimal quorum threshold, and cost asymmetry and optimal quorum threshold. However, because they did not apply signal detection theory to optimise individual agents' decisions, they were unable quantitatively to uncover the complex nonlinear relationship between these three quantities".

In fact, theorem 3.1 of the Ben-Yashar and Nitzan, 1997, paper gives the exact, analytical result of what the optimal quorum threshold is, as a function of the type I and type II error probabilities (p_i^1 and p_i^2 in that equation), the prior probabilities (α in that equation), and the costs and benefits of the possible outcomes (B in that equation). Rather than a qualitative result, this is an exact, analytical solution that tells you the optimal quorum threshold as a function of all of the variables that the authors of the present manuscript are interested in (it even goes further to allow for individual differences in the probabilities of committing type I and II errors, which was not considered in the present manuscript).

Indeed, it seems that the results in the present manuscript are actually the more qualitative of the two, since most of their results are in the form of inequalities, compared to the exact solution given by Ben-Yashar and Nitzan. The authors themselves suggest as much when they write "…it is possible to optimise individual quorums [Ben-Yashar and Nitzan, 1997] rather than simply set them within a suitable interval as we do here…".

The characterization that the Ben-Yashar and Nitzan paper provided only a qualitative solution, and the present manuscript a quantitative solution, therefore seems incorrect. One could in fact make an argument that the older paper is more precise, and goes further than the present one in several ways. It seems unfair to me for that paper to be mentioned for the first time only in paragraph one of the Discussion section (briefly), and then discussed in paragraph three in a couple of sentences. I feel that paper deserves a much more prominent role in the current manuscript, with its contributions, and the present manuscript's contributions (e.g., using signal detection theory, emphasizing certain errors), accurately characterized and compared.

2) The point regarding the Kao and Couzin, 2014 paper is of much lower importance than the above point, but since the authors disagreed with my point, I'd just like to clarify it, as I do think that there's a mapping between the two models. First, the authors mistakenly refer to one of the cues as the low 'accuracy' cue (and also refer to it as such in the manuscript in paragraph one of the Discussion), when it is actually a low 'correlation' cue -- this should be corrected. Furthermore, in the Kao and Couzin model, the probability rH that the high correlation cue gives correct information can be mapped to the prior probability p of being in state + in the present manuscript, since the state of the high correlation cue is a global one that affects all of the individuals equally. Then, if the high correlation cue gives 'correct' information, the probability that an individual selects the correct option (a+ in the current notation) is given by q*rL + (1-q) (in the notation of the Kao and Couzin paper, where I've changed p to q for clarity). On the other hand, if the high correlation cue gives incorrect information (a- in the current notation), then an individual selects the correct option with probability q*rL. Plugging these quantities into Equation 3 of the present manuscript gives rH*(1-q) + q*rL > 1/2, which must be true if rL > 1/2 and rH > 1/2, as was assumed in the Kao and Couzin paper (but q can vary from 0 to 1). The condition a+ > 1/2 then maps to q*(1-rL) < 1/2, which again must be true if rL > 1/2. Finally, the condition a- < 1/2 maps to q < 1/(2*rL), which is the condition shown in Figure 1B of the Kao and Couzin paper.

So the Error Ia condition does in fact map onto the boundary shown in Figure 1B of that paper. I concede that it's not a very obvious mapping, but it is there nonetheless. Indeed, the 'voting strategy' described in the Kao and Couzin paper could be interpreted as a 'soft,' probabilistic, quorum threshold, performed at the individual level, in contrast to the hard quorum threshold performed at a group level in the present manuscript.

---

## [Author Response]

Concerns:1) To "explain the error of our ways" is fine, expect some (if not most) of the theory and empirical work cited acknowledge that the Condorcet's theory represents a very simplified view of the world, and more complex models/mechanisms can be employed. We would like to see the authors 'soften' the critiques of earlier work so to better represent the advance being made here (e.g. paragraph one and two, Discussion section) and ensure researchers in the field take forward and use the results presented.

We have attempted to soften the language in some places in the passage highlighted by the editor, but how this reads will of course be subjective. However we have been able to acknowledge a positive instance of considering error asymmetry in the behavioural ecology literature, thanks to reviewer 1’s comments as discussed in the editor’s point 2 below, as well as a positive instance of observing group accuracy converging to intermediate values, thanks also to reviewer 1’s observations.

2) Regarding earlier work, reviewer #1 points to an earlier model that presents a similar model and conclusion to that presented here (Ben-Yashar and Nitzan, 1997). As a behavioural ecologist who has interest in these models, I was unaware of it, and therefore many in the field may be in the same position. As the work is presented in a narrative form,… "first we did this, next we tested, etc." it would be useful to weave into the narrative how this work is similar or different to what is being presented, as explained nicely by reviewer #1. Moreover, the 1970s through 1990s saw a lot of theoretical extensions to Condorcet, and today, there's a lot of work studying the effect of social influence, correlations, heterogeneity, emergent sensing, etc. on collective wisdom, which are all deviations from Condorcet. This should be more explicitly acknowledged early in the manuscript.

We agree on the importance of the paper the reviewer has very helpfully pointed us towards, which we were unaware of. We have rewritten portions of the paper to clarify the relationship of our results to those of Ben-Yashar & Nitzan. Briefly, while Ben-Yashar & Nitzan indeed present the general solution and note the relationship between asymmetries of payoffs and priors, and interestingly unify confidence weighting with quorum rules, they do not apply signal detection theory to model individual decision-makers (beyond assuming the basic relationship between true positive and false-positive rates). Because of this, while they are able to make qualitative discussions of the relationships between asymmetries and the resulting changes in quorum threshold, they cannot quantify this, or reason about interactions. Our quantitative approach does enable this and reveals such interactions to be non-linear (Figure 2), however, and is, based on our application of the mathematics of signal detection theory, our primary contribution. Awareness of this paper has also helped us appreciate where existing behavioural ecology research *has* realised the importance of quorums in this situation, and acknowledge it appropriately, enabling us better to respond to the editor’s point 1, as described above.

We do however feel we explicitly acknowledge existing work on generalising Condorcet’s result, citing a number of relevant papers in the following passage:

“Over the past few decades, substantial research effort has focussed on two key explicit assumptions underlying Condorcet’s Jury Theorem, independence (i.e. judgments/votes by different members of the group are assumed to be independent from each other) and homogeneity (i.e. all decision-makers within a group are assumed to be identical, both in competence and in goals) [12, 16, 22-25].”

3) Related to points 1 and 2, the paragraph setting out the three key interpretations (Introduction section) requires citations to back up each of the statements. I can think of one of my own papers that would be appropriate to cite, but how many more? The authors are suggesting we have all been misled, so we need to have a good number of citations here to back up this claim.

We have included these references as requested.

4) The definition of 'optimal' and its relation to accuracy in decision-making (here, related to probability of correct decision) needs to be explained properly in the Introduction, or else a different more representative term used (accurate/accuracy?).

In the first occurrence of optimal outside the Abstract we have clarified our definition. In the Discussion we have also clarified that it is possible to optimise the quorum threshold used, and explained why that is not our approach here.

5) Non-independence of errors is probably the rule rather than the exception in nature, as well as in lab experiments with animals or humans. Both reviewers query if and how this will effect either the construction of "state-of-the-art" models like this one, or their interpretation and generality – if not explicitly incorporated. At very least, this should be discussed.

We have addressed this in the closing Discussion, in also introducing Bayesianism as an approach (editor’s point 6 and reviewer 2’s points below).

6) Reviewer #2 thinks a Bayesian analysis would be as or more insightful than the signal detection theory, then it is likely so will others. Therefore, I would encourage Authors to present an argument against this in the discussion (where the use of signal detection theory is highlighted: paragraph three, Discussion section), and/or consider their recommendation to work out a general solution/expression using Bayesian analysis.

We now cite such approaches in the Discussion and argue that our approach has the merit of simplicity, and revealing interesting structure of collective decisions even with very stringent assumptions. We feel that conducting such an analysis in this revision would result in a substantially different manuscript, so are grateful for the editor’s and reviewer 2’s willingness for us to justify not doing this. We are, however, committed Bayesians and hope future work may apply this approach further to the scenario we have considered here.

7) The Abstract should begin with one or two sentences giving a broader background. At present, it begins with very specific background regarding the Condorcet Jury Theorem. Also, the biological system should be indicated in the title and/or Abstract: please revise the title and/or Abstract with this advice in mind. A simple solution would be to could add "in biological systems" (or similar) to the title and then explain in the Abstract that the theory/models apply to all sorts of 'social' organisms.

Unfortunately the existing Abstract gives us only 6 more words to work with; feeling unable to jettison any of the existing Abstract text we have added “Collective decision-making is ubiquitous” to the beginning of the Abstract – we have also added ‘in biological systems’ to the title, as suggested.

Separate reviews (please respond to each point):

Reviewer #1:

[…] I agree with the authors that these two ideas need to be of greater theoretical and empirical research, and this manuscript does a good job of highlighting the importance of these ideas. The authors rightly show that in nature, the prior probabilities, and the costs/benefits, of different possible outcomes are generally not equal (there may rarely be a predator, but when there is one, the cost of being eaten is very high). They link the large literature on quorum decision-making to optimal decisions, demonstrating how quorum decisions may be optimal within the ecological scenarios in which species evolve, which is important.My main concern with this manuscript is its novelty. In particular, there are two previously published papers (one which the authors cited and one which the authors did not), which together contain all of the main ideas and much of the results of the present manuscript. I describe these papers below, and their intersection with this manuscript, and then describe what I think are the novel sections remaining in this manuscript.The first paper, which the authors did not cite, is: Ben-Yashar RC and Nitzan SI (1997) The optimal decision rule for fixed-size committees in dichotomous choice situations: the general result. International Economic Review, 38(1):175-186. The model in this paper incorporates the main ideas of the present manuscript, namely that the costs/benefits ('payoffs') may be different for different outcomes; the prior probability of different outcomes may be different; and the probabilities of false positives and false negatives may be different. The model described in Ben-Yashar and Nitzan, 1997, is essentially identical to the extended model described in the current manuscript. That paper also showed that 'quorum' rules are generally optimal, but rather than denote the region of parameter space in which this is the case, those authors went further and mathematically proved the exact value of the quorum threshold for each set of parameter values (they also further considered heterogeneous individuals, which was not considered in the current manuscript). In short, the optimal solution for the model appears to have been solved already.

We thank the reviewer for drawing this very important paper to our attention, which we now cite and discuss as described in our response to the editor’s point 2 above; in that response we also discuss how our results are still novel in this revised context.

The second paper, which the authors did cite, is: Kao and Couzin, 2014. This paper used a slightly different model but with some key similarities. In particular, the model described two cues in the environment, one which is independently perceived by each individual, and one which is globally perceived by all individuals. Although the authors did not describe the global cue as such, one interpretation of a global cue is as a prior probability of an outcome. By mapping the model in the current manuscript to the Kao and Couzin, 2014 model, one can find that the condition for "Error Ia" is the same as the boundary shown in Figure 1B of the Kao and Couzin paper. In the latter paper, the authors show that in this region of parameter space, the collective accuracy for very large groups approaches the accuracy of the global cue, which is the same result as Equation 5 and the left column of Figure 3 in the present manuscript.

Again, we thank the reviewer for drawing this aspect of the paper we cited to our attention; however we respectfully disagree that the result maps onto our Error Ia; in Kao and Couzin’s model individuals attend to one cue or another with a probability determined by their individual strategy — we do not see a formal analogy with our model in this regard, and feel that the nature of Figure 1B compared to our Figure 2A, in which plot axes represent different quantities, highlights the difference between the results — we also do not see that the low accuracy cue in Kao and Couzin is analogous to the prior in our model; in Kao and Couzin individuals have a propensity to attend to the low accuracy cue, whereas in our model the prior leads to a shift in the optimal quorum threshold. However we do agree that the *reason* for Kao and Couzin’s result in their Figure 1B, and our error Ia, are the same, and now cite their paper in that additional context as well as comparing the two models in our revised Discussion.

To summarize, the model described in the present manuscript appears nearly identical to the one described in the Ben-Yashar and Nitzan paper, while the older paper gave a more general solution to than the present manuscript. Furthermore, one of the three errors presented in the present manuscript has been essentially described in the Kao and Couzin paper.Despite all of this, I do think that there remain some novel results in the present manuscript. To my knowledge, Error Ib and Error II have not been previously described, although Error Ib is a subset of Error Ia. Furthermore, the way the authors parsed out the different errors and illustrated its prevalence in parameter space is useful. I also think that despite this model being solved already, the Ben-Yashar and Nitzan paper is not broadly known to the ecology/behavior community and drawing attention to it and its results is important. However, I think that the authors need to revise their manuscript in light of these two papers to highlight the aspects of their work that are truly novel.

Once again we thank the reviewer for pointing out these links to pre-existing work, and for the reasons discussed above we still consider that our results are substantially novel, in which we are in agreement with the reviewer we believe.

Minor Comments:I would suggest that the authors move most, or all, of the appendix to the main text, as it's not possible to understand the model or the results without reading this. For example, it's not clear why there is a trade-off between a+ and a- without knowing the details of the model (one may wonder why one can't simply set a+ and a- both to 1).

We didn’t feel this would aid the flow of the manuscript, as it would represent a substantial digression at that point in the manuscript; when the appendix is first discussed (before Equation 2) it is made clear that the appendix explains how pairs of a+, a- values are determined. Additionally, for some readers the signal detection theory appendix may be redundant, as signal detection theory is a staple of many behavioural ecology textbooks, such as the one we cite in our manuscript.

Reviewer #2:The authors extend Condorcet analysis by taking into account false and negative errors instead of collapsing them into a single error. They work out the consequences of their analysis. I completely agree with the authors that the role of Condorcet theorem has been misinterpreted in the literature. It is a nice analytical model full of assumptions and we can learn a lot by realizing what is the role of these assumptions.The author's key insight is that the world, for the agent, can be in different states, say + and -, with probabilities p and 1-p, and the error in each world state is different. In contrast, Condorcet's treatment is for a single world state.There are two ideas I would consider:1) I think a Bayesian analysis is actually as or more insightful than the signal detection theory. It naturally takes into account all types of errors and even correlations. You can also see in this treatment that you can do very well in supermajority or below majority cases.

We simultaneously agree and disagree — in our revised Discussion we now justify the utility of our simpler analysis, with more assumptions, as described in our response to the editor’s point 6.

2) In this respect, the analysis in the paper assumes errors to be independent. This is fine, Condorcet assumes independent voters. But this assumption does not hold in practise. So, again, I think the more general treatment is a Bayesian one including all types of errors and correlations (see Perez-Escudero and Polavieja, 2011 and more recently in Interface).I think it would be nice to work it out from Bayes (maybe even more general using costs), obtain the general expression, and work out the signal-detection-theory case as a particular case.As you are in this new experimental reviewing mode, you might try my suggestion (or not), and if it gets messy, leave it for another time (or not).

We thank the reviewer for noting that this may be more work than is appropriate for a manuscript revision. In our opinion this would substantially change the nature of our manuscript, and therefore now highlight the interest in following a Bayesian approach in future work, as well as explaining the interest in our simpler approach.

[Editors' note: further revisions were requested prior to acceptance, as described below.]

Reviewer 1 comments:1) Regarding the Ben-Yashar and Nitzan, 1997, paper, the authors write in their response letter: "…while they are able to make qualitative discussions of the relationships between asymmetries and the resulting changes in quorum threshold, they cannot quantify this, or reason about interactions," and in their revised manuscript, "These results describe a qualitative relationship between prior asymmetry and optimal quorum threshold, and cost asymmetry and optimal quorum threshold. However, because they did not apply signal detection theory to optimise individual agents' decisions, they were unable quantitatively to uncover the complex nonlinear relationship between these three quantities".In fact, theorem 3.1 of the Ben-Yashar and Nitzan, 1997, paper gives the exact, analytical result of what the optimal quorum threshold is, as a function of the type I and type II error probabilities (p_i^1 and p_i^2 in that equation), the prior probabilities (α in that equation), and the costs and benefits of the possible outcomes (B in that equation). Rather than a qualitative result, this is an exact, analytical solution that tells you the optimal quorum threshold as a function of all of the variables that the authors of the present manuscript are interested in (it even goes further to allow for individual differences in the probabilities of committing type I and II errors, which was not considered in the present manuscript).Indeed, it seems that the results in the present manuscript are actually the more qualitative of the two, since most of their results are in the form of inequalities, compared to the exact solution given by Ben-Yashar and Nitzan. The authors themselves suggest as much when they write "…it is possible to optimise individual quorums [Ben-Yashar and Nitzan, 1997] rather than simply set them within a suitable interval as we do here…".The characterization that the Ben-Yashar and Nitzan paper provided only a qualitative solution, and the present manuscript a quantitative solution, therefore seems incorrect. One could in fact make an argument that the older paper is more precise, and goes further than the present one in several ways. It seems unfair to me for that paper to be mentioned for the first time only in paragraph one of the Discussion section (briefly), and then discussed in paragraph three in a couple of sentences. I feel that paper deserves a much more prominent role in the current manuscript, with its contributions, and the present manuscript's contributions (e.g., using signal detection theory, emphasizing certain errors), accurately characterized and compared.

We thank the reviewer for their further comments on the relationship between Ben-Yashar and Nitzan’s work, and our own, and thank them again for highlighting this very important paper in their initial review. We fully agree with the reviewer’s suggestion that the work be cited and discussed earlier in the text given its relevance and we now do exactly this in the Introduction of our paper. Also, as a response to the reviewer’s comment, we further adjusted our discussion of Ben-Yashar and Nitzan in the Discussion section of our paper – in particular, we followed the reviewers suggestion and again highlight the importance of Ben-Yashar and Nitzan and do not refer to their results as qualitative any more. To summarise our argument here, since Ben-Yashar and Nitzan did not relate true positive rate and false positive rate to prior and cost asymmetries, they were unable to quantitatively relate optimal quorum threshold to use of a super or sub-majority quorum, as we do here for the first time.

2) The point regarding the Kao and Couzin, 2014 paper is of much lower importance than the above point, but since the authors disagreed with my point, I'd just like to clarify it, as I do think that there's a mapping between the two models. First, the authors mistakenly refer to one of the cues as the low 'accuracy' cue (and also refer to it as such in the manuscript in paragraph one of the Discussion), when it is actually a low 'correlation' cue -- this should be corrected. Furthermore, in the Kao and Couzin model, the probability rH that the high correlation cue gives correct information can be mapped to the prior probability p of being in state + in the present manuscript, since the state of the high correlation cue is a global one that affects all of the individuals equally. Then, if the high correlation cue gives 'correct' information, the probability that an individual selects the correct option (a+ in the current notation) is given by q*rL + (1-q) (in the notation of the Kao and Couzin paper, where I've changed p to q for clarity). On the other hand, if the high correlation cue gives incorrect information (a- in the current notation), then an individual selects the correct option with probability q*rL. Plugging these quantities into Equation 3 of the present manuscript gives rH*(1-q) + q*rL > 1/2, which must be true if rL > 1/2 and rH > 1/2, as was assumed in the Kao and Couzin paper (but q can vary from 0 to 1). The condition a+ > 1/2 then maps to q*(1-rL) < 1/2, which again must be true if rL > 1/2. Finally, the condition a- < 1/2 maps to q < 1/(2*rL), which is the condition shown in Figure 1B of the Kao and Couzin paper.So the Error Ia condition does in fact map onto the boundary shown in Figure 1B of that paper. I concede that it's not a very obvious mapping, but it is there nonetheless. Indeed, the 'voting strategy' described in the Kao and Couzin paper could be interpreted as a 'soft,' probabilistic, quorum threshold, performed at the individual level, in contrast to the hard quorum threshold performed at a group level in the present manuscript.

We thank the reviewer for pointing out our mislabelling of Kao and Couzin’s variables and corrected this error, that is, the cue that we mistakenly labeled as “low accuracy cue” is now labeled as “low correlation cue”. We also thank the reviewer for further explaining their reasoning about analogies between our analysis, and that of Kao and Couzin. However we maintain our position that while the results have the same mechanism at their heart, they are not equivalent. The fundamental question is (from the initial review) whether “by mapping the model in the current manuscript to the Kao and Couzin, 2014, model, one can find that the condition for 'Error Ia' is the same as the boundary shown in Figure 1B of the Kao and Couzin paper". As we argued before, a high correlation cue, despite being accessible to all individuals in Kao and Couzin’s model, is not equivalent to a prior – individuals do not directly choose to attend to a prior or not, whereas they do in Kao and Couzin. In response, the reviewer claims to have shown how our inequality 3 can explain their (Kao and Couzin’s) boundary in Figure 1B, through deriving our quantities a_+ and a_- in terms of the their variables. That the two results are not the same can be seen most easily by observing that a_+ and a_- are related to each other by a simple linear relationship according to the reviewer’s derivation, which is a function of the individual agent’s strategy and a simple accuracy (i.e. neglecting the possibility of two error types, which is fundamental in our analysis); in contrast, in the result presented in the manuscript these quantities are independent, bar the simple relationship a_+ > 1-a_-. Therefore Kao and Couzin’s condition might best be seen as an instantiation of our condition in a special case. However, even then we dispute the equivalence since, as discussed above, the two decision problems are quite different. While we do not disagree with the reviewer’s symbol manipulation, it is not surprising if models of different decision problems give comparable mathematical expressions when dealing with expectations, and sharing a few similarities such as majority rule. We therefore maintain that the reason for Kao and Couzin’s result is the same as the reason for ours, as we already wrote in the revised manuscript, and as the reviewer writes in their subsequent comments. However we feel that providing further argumentation about perceived similarities between the results is not illuminating and in fact risks confusion. Therefore we have not revised our manuscript further in this regard, except to correct the aforementioned error in labelling Kao and Couzin’s variables.